# Synthesis and Characterization of Functionalized Chitosan Nanoparticles with Pyrimidine Derivative for Enhancing Ion Sorption and Application for Removal of Contaminants

**DOI:** 10.3390/ma15134676

**Published:** 2022-07-03

**Authors:** Mohammed F. Hamza, Yuezhou Wei, Khalid Althumayri, Amr Fouda, Nora A. Hamad

**Affiliations:** 1School of Nuclear Science and Technology, University of South China, Hengyang 421001, China; 2Nuclear Materials Authority, P.O. Box 530, El-Maadi, Cairo 11728, Egypt; 3School of Nuclear Science and Engineering, Shanghai Jiao Tong University, Shanghai 200240, China; 4Department of Chemistry, College of Science, Taibah University, Al-Madinah Al-Munawarah 30002, Saudi Arabia; kthumairi@taibahu.edu.sa; 5Botany and Microbiology Department, Faculty of Science, Al-Azhar University, Nasr City, Cairo 11884, Egypt; amr_fh83@azhar.edu.eg; 6Chemistry Department, Faculty of Science, Menofia University, Shebin El-Kom 32511, Egypt; nhamad059@gmail.com

**Keywords:** modified chitosan, chromium removal, uptake kinetics, pyrimidinedione derivative, eco-friendly adsorbent

## Abstract

Modified chitosan has been widely used for heavy metals removal during the last few decades. In this research, the study was focused on the effect of modified chitosan particles after grafting with heterocyclic constituent for enhancing the sorption of Cr(VI) ions. Chitosan was functionalized by 2-thioxodihydropyrimidine-4,6(1H,5H)-dione, in which the synthesized composite considered as a nanoscale size with average 5–7 nm. This explains the fast kinetics of sorption with large surface area. The prepared sorbent was characterized by Fourier-transform infrared (FTIR), elemental analysis (EA), Brunauer–Emmett–Teller (BET surface area) theory, thermogravimetric analysis (TGA), mass spectroscopy, and scanning electron microscopy (SEM) with energy dispersive X-ray analysis (EDX) analyses. The experimental part of this work involved the application of the synthesized sorbent for the removal of Cr(VI) ions from highly contaminated tannery effluents that are characterized by a high concentration toward chromate ions with other associated toxic elements, i.e., Pb(II) and Cd (II) ions, which underscore the importance of this treatment. Under the selected conditions (K_2_Cr_2_O_7_ salt, C_o_: 100 mg L^−1^ and pH: 4), the sorption diagram shows high Cr(VI) sorption and fast uptake kinetics. The sorption was enhanced by functionalization to 5.7 mmol Cr g^−1^ as well as fast uptake kinetics; 30 min is sufficient for total sorption compared with 1.97 mmol Cr g^−1^ and 60 min for the non-grafted sorbent. The Langmuir and Sips equations were fitted for the sorption isotherms, while the pseudo-first order rate equation (PFORE) was fitted for the uptake kinetics.

## 1. Introduction

As result of our current industrialized life, environmental contamination has become an important global problem [1,2]. Among the running contaminants, the pollution of the environment by heavy elements is a critical calamity. Heavy metals generally result from different industries such as mining, batteries, metal lamination, etc. [3,4,5,6]. Heavy metals are elements characterized by their relatively high molecular weights and densities. These elements are non-biodegradable and have more accumulative behavior than other organic pollutants in living organisms. So they are dangerous, toxic and carcinogenic elements [7,8]. The most common heavy element water pollutants are the cadmium, copper, chromium, zinc and lead [9,10]. Chromium in turn is considered one of the most dangerous water pollutants. Because of the toxic properties of chromium and its dangerous environmental problems, there were a persistent need to perform many studies in order to find the best approaches for the removal of Cr(VI) [11].

Chromium is widely used in the electroplating, textile, leather, metallurgy, tanneries, metals industries and metal finishing [12]. Such types of industries involve the emission of huge amounts of outflowing to the outer environment carrying various chromium ions species with high levels. These released ions pollute surface and groundwater as well [13,14,15,16,17]. The treatment of such contaminated water/effluents is a critical point in these industries, to minimize their impact toward water bodies and environment [18]. Various techniques have been used for this mission, including electrochemical reduction [19], precipitation [20] (after reduction of Cr(VI) to Cr(III) [21]) or bio-reduction [22]. The acidic medium enhances the reduction of Cr(VI) to Cr(III) for easily precipitation. Solvent extraction [23,24] and the impregnated process (solvent-impregnated resins) are also used for the recovering and of chromate anions [25,26,27]. Other studies are based on using zero-valent iron, iron oxide, and magnetite sorbents [28,29] for the removal of chromate anions. Chromium also has a dangerous toxic and carcinogenic behavior [30]. The standard allowed limit of chromium ions in drinking water was shown by the WHO in 2008. In case of exceeding that limit (0.05 mg L^−1^), chromium ions must be removed from wastewater. Recently, researchers have focused on developing eco-friendly and low-cost adsorbents for removing Cr(VI) [31,32,33], in which the efficient uses of the green synthesized nanoparticles in removal of heavy elements and Cr ions [34,35]. In this regard, chitosan modification by various chemical (by grafting of new functional groups) and physical methods (fabrication as beads or gels) has played an important role as an effective technique to remove inorganic and organic contaminant of aqueous intermediates [36,37]. Chitosan is a natural functionalized biodegradable polymer produced from the partial deacetylation of chitin. Having -OH and -NH_2_ groups gives it a characteristic structure which in turn contributes its unique adsorption properties [38,39,40,41]. The phyco-chemical properties of chitosan particles depends on the deacetylation degree, molecular weight but also the solubility [42,43,44]. Due to the high nitrogen percent in its structure compared with synthetic cellulose derivatives, chitosan has an important commercial interest [45,46]. In recent decades, many studies were performed to improve the solubility of the modified chitosan structures [47]. Cross-linking of chitosan by chemical methods affects its physicochemical properties, and as a result, applications are also affected due to the irreversible gelation process. Covalent cross-linking also increases the stability (chemical and thermal stability) of modified derivatives of chitosan. Thus, chemical cross-linking is widely applied by aldehydes and epoxy such as glutaraldehyde, glyoxal and epichlorohydrin [48,49,50,51,52,53,54].

Glutaraldehyde is a widely used crosslinker in modification of chitosan in adsorption process. Carboxymethyl chitosan (cmc) cross-linked with glutaraldehyde was prepared to modulate the selectivity of modified chitosan derivatives for heavy metals adsorption [55]. Among the most common crosslinkers epichlorohydrin (epi) is widely used to synthesize modified chitosan sorbents, which are mainly applied in removing wastewater pollutants [56,57]. Many studies have been published during the last few years on the different applications of chemically modified chitosan composite, especially in water and wastewater treatment. In this respect, heavy metals removal is considered a big challenge for many scientists all over the world. Cr(VI) removal gained most of the interest among other heavy metals because of its well-known dangerous effect on the total environment [58,59,60].

This work has many tasks: first, synthesis of the grafted moieties through cyclization reaction of diethyl malonate and thiourea to the pyrimidine derivatives; second, synthesis of chitosan nanoparticles and activated chloride spacer arm for possible reaction with the pyrimidine moiety to yield the MC-TDP nanoparticles. Then, sorption properties of the produced grafted composite (compared with the nongrafted chitosan (MCH) as reference material) for optimization of the sorption properties, which indicated an improvement in the sorption behavior, uptake kinetics and also the cycling properties. The final stage concern the application toward nature contaminated effluents from tannery industries, which verify the valorization of the synthesis by removing more than 99% of Cr(VI) using MC-TDP compared to less than 30% by MCH under the same condition of pH, loading time and sorbent dose. The sorbent was characterized by mass spectroscopy (for the TDP particles), FTIR, SEM, SEM EDX, TGA and BET (surface area) through N_2_ sorption–desorption isotherms. The sorption was performed twice for the reproducibility properties.

## 2. Materials and Methods

### 2.1. Materials

Ferrous sulfate, ammonium ferric (III) sulfate dodecahydrate ((FeSO_4_⋅7H_2_O; >99.0%) and (NH_4_)Fe(SO_4_)_2_⋅12H_2_O; >99.0%, respectively), epichlorohydrin (epi; >99.0%), chitosan (Degree of Acetylation (DA): greater than 75%) (CHI), sodium hydroxide pellets (>99.99% for pH adjustment), sodium ethoxide (C_2_H_5_ONa) solution (50%), thiourea (CH_4_N_2_S; 99%), diethyl malonate (C_7_H_12_O_4_; 99%), glutaraldehyde (C_5_H_8_O_2_; 25% in H_2_O) and potassium dichromate (K_2_Cr_2_O_7_) were supplied by Sigma Aldrich (Merck, Darmstadt, Germany). Acetone (99.9%) and dimethylformamide (DMF; 99.9%) were purchased from Chron Chemicals (Qionglai, China). Other chemicals were obtained from Prolabo (VWR, Avantor Group, Fontenay-sous-Bois, France), which are used without further purification.

### 2.2. Preparation of the Sorbent

#### 2.2.1. Synthesis of Thioxodihydropyrimidine Dione Derivative (TDP)

To a warmed sodium ethoxide solution 50% (20 mL), thiourea (0.64 g) and diethyl malonate (1.5 g) were added drop wise. The mixture was kept at reflux temperature (80 °C) for around 10 (±1) h, then poured onto the ice beaker in 0.1 M HCl medium and left overnight. The prepared solid product underwent separation by filtration, dried, and recrystallized from ethanolic solution to give TDP as a white powder (80% yield), m.p. above 300 °C, as shown in Appendix A.

#### 2.2.2. Synthesis of Nano-Magnetite Activated Spacer Arm Composite

The magnetite nanoparticles was firstly prepared by thermally precipitation procedure [61]. Dissolving of 7.35 g ammonium iron (III) sulfate and 5.0 g of the hydrated iron (II) sulfate in distilled water at 50 °C, the reaction mixture was continued under stirring for 60 min. The pH of the mixture was adjusted to value 11 using NaOH solution, this process for magnetite precipitation, followed by continue heated at 50 °C for 5 h. Then, it was magnetically separated and washed with water, then dried at 50 °C overnight.

The second step of the synthesis is the preparation of chitosan magnetite nanoparticles through dissolving 4 g of chitosan particles in 100 mL of diluted acetic acid solution (7%), then addition of the prepared magnetite (4 g) powder with continuous stirring. The mixture was precipitated by adjusting the pH to 10 using NaOH solution. The precipitated particles were heated to 90 °C for 2 h. After collection of magnetite chitosan particles, the next step concerning with enhancing the stability by crosslinking properties by addition of 0.01 M epi in basic medium of NaOH solution (the pH was adjusted to 10) and reaction maintained stirring for 3 h at 60 °C. After magnetically separation, the further addition of epi was performed in ethanolic solution (1:1 ethanol/water) for creation of activated methylene chloride as a spacer arm through addition of 18 mL of epi and the mixture was refluxed at 70 °C for 3 h. The produced activation moiety was magnetically separated and washed with acetone and water then dried at 60 °C for 12 h, as shown in Appendix A.

#### 2.2.3. Synthesis of Grafted Pyrimidine on Chitosan Magnetite Nanoparticles (MC-TDP)

To 150 mL of DMF solution, 5 g of TDP was added for dissolution, followed by addition of 6 g of Mg-chitosan with gentle stirring. The pH of the mixture was adjusted to 9.5 using sodium hydroxide solution. The mixture was refluxed for 9 h at 85 °C, followed by stirring for 24 h. The product was filtered and washed with distilled water, then it was dried overnight at 50 °C, Appendix A. Appendix A reports the cost evaluation of 10 g modified grafted sorbent.

### 2.3. Characterization of Materials

FT-IR spectra were identified by IRTracer-100 FT-IR, spectrometer, Shimadzu, Tokyo, Japan; the dried samples were mixed and grinded with KBr. The surface area (SA) and porosity of the prepared sorbents were applied under nitrogen atmosphere for adsorption/desorption isotherms through Micromeritics (Norcross, GA, USA), TriStar II-Norcross-GA; USA-system-77-K. The BET equation was applied for adsorption, after sweeping of the sorbent samples in nitrogen for around 7 h at 130 (±10) °C. The morphology structure of the samples was characterized by SEM analysis using Phenom-ProX-SEM (Thermo Fisher Scientific, Eindhoven, The Netherlands). The element contents before and after metal adsorption, was semi-quantitatively determination using EDX-analysis (i.e., energy dispersive X-ray analysis). The pH_pzc_ (zero charge) was measured using the drift method [62]. The thermal determination was studied via TGA-analysis using the Netzsch STA-449F3 Jupiter, NETZSCH-Gerätebau-HGmbh, Selb, Germany. The test was achieved under nitrogen atmosphere media at 10 °C min^−1^ (described as the temperature ramp). The pH adjustment of the tested solution was performed using a pH-26A, Acculab USA pH-meter (Acculab USA, New York, NY, USA). The collected samples were firstly filtrated (using 1.2-µm filter membranes) before testing for metal content. The chromium ions were measured (before and after adsorption) by ICPS-7510-Shimadzu, Tokyo, Japan. The morphology as well as the particle size measurements were analyzed using TEM analysis (transmission electron microscopy); JEOL, 1010-JEOL Ltd., Tokyo, Japan.

### 2.4. Sorption Properties

Sorption efficiency was studied in batch closed systems at room temperature, a fixed amount of sorbent (m, g) was mixed with a fixed volume of solution (V, L) under agitation conditions of v: 210–215 rpm. The used dose of sorbent (SD) in most cases was adjusted to 0.6 (±0.1) g L^−1^. The effect of pH was studied by using a pH-26A Acculab USA pH-meter (Acculab USA, New York, NY, USA). The initial pH values (i.e., pH_in_) varied between 1 and 6; the pH was not controlled during the sorption, but the final pH (i.e., pH_eq_) was recorded. The contact time (t) of the experiments (i.e., pH effects test, uptake kinetics, selectivity (equi-polymetallic) test, and sorption isotherms, etc.) were investigated (at fixed period) at 24 h. In the sorption isotherms and the uptake kinetic tests, the pH_in_ was fixed at value 4. The initial concentrations (C_0_; mmol Cr(VI) L^−1^) were varied for sorption isotherms (0.01 to 6.1 mmol Cr L^−1^), while the selectivity tests was investigated with tannery effluents containing high concentration of associated elements with regard to the WHO regulation (as Ni(II), Pb(II), Co(II), and Cd(II)) at pH_0_: 4.01. The capacity (q_eq_, mmol g^−1^) of the bound metal ions was determined from (mass balance) equation: q_eq_ = (C_0_ − C_eq_) × V/m, in which the D, L g^−1^ (distribution ratio)was calculated by the following equation: D = q_eq_/C_eq_. The desorption kinetics were determined for the chromium-loaded sorbent (collected from the kinetic test) using diluted HCl solution (0.2 M). The models used in the kinetics and isotherms were described with the parameters listed in Appendix A.

### 2.5. Application to the Tannery Waste Effluent

The waste effluent was collected from the Robbiki Leather (10th of Ramadan City) in Cairo, Egypt. The GPS information was assigned as follows: N: 30°17′898″, E: 31°76′840″. Sorption was tested in batch reactor with pH_in_ 4.01 for different sorbent doses (SD; 1–30 g L^−1^) for 24 h of contact time.

## 3. Results

### 3.1. Characterization of Sorbents

#### 3.1.1. FTIR Analysis

The chemical structure elucidation and the new functional groups of the modified sorbent of the TDP, MC-TDP and the chemical modifications/changes occurred as Cr(VI) sorption, and after 5 cycles of sorption and desorption, they were investigated by the FTIR spectroscopy (Figure 1 and Appendix A). The TDP moiety characteristic peaks (cm^−1^ ν max) were identified as follows: 3463–3382 cm^−1^ (νNH_2_ and OH) overlapping stretching band [50], 2871 cm^−1^ (νCH_2_) stretching, and 1625–1705 cm^−1^ (νC=O and C=C) stretching of the heterocyclic ring. This illustrates the cyclization of the heterocyclic moiety after the reaction of thiourea and diethyl malonate. The bands at 2673 cm^−1^ and 1611 cm^−1^ are assigned for (νSH) stretching of thiol groups (after tautomerization with amine groups) [5], and the stretching vibration of amide (νC=N) (produced from tautomerization rearrangement), overlapped with N-H bend [63], respectively, and 1524 cm^−1^ for (νC=S) stretching. Peaks that appeared in the range of 727–627 are related to the (NH_2_ + CO) wag. vibration [64] and N-C-N bend (in-plane), and this band is overlapped with the magnetite (Fe-O) [65]. As modifications were performed, most of these peaks not only changed in intensity or shifted to new wavenumber but also new peaks appeared related to this modification and the new reactive groups. Decreasing intensity of NH stretching for TDP with two pikes at 3584 and 3392 cm^−1^ was noticed, and the broadness was increased again as a result of grafting (appearing at 3443 cm^−1^ for the MC-TDP, which related to the high percent of amine groups). The C-H stretching related the aromatic heterocyclic moiety appeared at 3095 cm^−1^ and then disappeared (overlapping) with the increasing broadness of OH, NH for the MC-TDP. Peaks at 1155 cm^−1^ (and 1097 and 1060) for TDP and MC-TDP sorbents are related to the NH_2_ (rock.), N-C str., C-O str. and C-O-C. (str., asymm) for hydrocarbon backbones [6]. The bands at 923 and 854 cm^−1^ (for TDP and MC-TDP, respectively) are related to the C-H (aromatic bend) for the pyrimidine ring (in-plane). After the sorption of chromate ions, it is noticed that most of assigned peaks are shifted with decreasing intensities; these changes are specially for groups used in binding with metal ions, i.e., amine (NH), carbonyl(C=O), thion (C=S) and thiols (SH) (resulting by tautomerization) [5,64,66,67], while as desorption processes were performed (5th cycles), these peaks were noticed again with full intensities as in the original sorbent (as shown in Figure 1), while the most important assigned bands are reported in Appendix A. The EI-MS shows *m*/*z* (C_4_H_4_N_2_O_2_S) calc. = 144 [M^+^]. These analysis charts are shown in Figure 2, while Appendix A shows the completed data of the mass analysis.

#### 3.1.2. Morphology and Textural Properties

SEM and TEM graphs of MCH and MC-TDP are shown in Appendix A. From the SEM micrograph in Appendix A, the sorbents were prepared as fine flaked particles with irregular shapes, and the average size is around 5–10 µm, while in Appendix A, the TEM micrograph illustrates the nanoscale size of the magnetite embedded into the polymer chain; these nanoparticles appear as heavy dark round dots with orderly shape. The magnetite clot caused the appearance of dark condensed areas in some parts in the matrix, while the size is ranged between 2–5 nm. Appendix A shows the attraction of functionalized magnetite chitosan to the external magnet, which verify the effect of magnetism.

The S_BET_ and the V_p_, specific surface area and the porous volume, respectively, of the prepared sorbents were measured by using the nitrogen sorption and desorption isotherms. These values are reported for the MCH sorbent to be 22.9 m^2^ g^−1^ and 6.8 cm^3^ STP g^−1^, respectively, and 24.1 m^2^ g^−1^ and 8.2 cm^3^ STP g^−1^ for MC-TDP, respectively; these parameters are noticeably affected by the modification procedures.

#### 3.1.3. Thermogravimetric Analysis

Figure 3 shows the thermally decomposition analysis of MC-TDP sorbent. Three decomposition stages were appeared, the first decomposition stage for the adsorbed water loss (at 270.8 °C); with about 12.09% loss percent, the second loss profile was related to polymer backbone depolymerization, amine degradation, and also for the carbonyl(C=O)/thiocarbonyl(C=S) decompositions; this step ranged to about ~23.1% of total loss and ranged from 270.8 to 375.5 °C. The final profile of decomposition was recorded in the range 375.5 to 496.7 °C, which was assigned to the char formation. The overall loss (totally) of the MC-TDP material is close to 57.038%. On the other hand, the DrTGA has five loss stages (57.56 °C, 290.6 °C, 399.9 °C, 595.6 °C and 761.2 °C) which differ in intensity, as shown in Figure 3. On the other hand, the sorbent was subjected to burning in a vacuum oven, at 900 °C for two hours, and the total loss is around 58.1%, which is close to that obtained from the TGA analysis.

#### 3.1.4. EDX and Elemental Analyses

Figure 4 represents the semiquantitative EDX analysis for both sorbents (MCH and MC-TDP). The nonmodified sorbent shows a lower percent of O and N than that of grafted heterocyclic sorbent; on the other hand, the sulfur that appeared in the grafted sorbent confirms the successive grafting of the TDP moiety. It was noteworthy that the presence of Fe ions verifies the stability and nondegradability of magnetite during synthesis, while also decreasing the Fe% in the MC-TDP more than that measured in MCH, which is due to the relative increase in the hydrocarbon constituents. The elemental analysis data is shown in Table 1, which illustrates the increase in N and O contents for the MC-TDP composite (5.63 and 32.96% for MCH-TDP, respectively), compared to the MCH composite (MCH; 3.98 and 27.14%, respectively), while sulfur appeared with 1.11% in MC-TDP, which in turn proves the successive grafting of the TDP substrates.

#### 3.1.5. Surface Charge Analysis—pH_PZC_

As shown in the synthetic procedures (Appendix A), the combination of heterocyclic moieties at the chitosan surface changes in the amine environment (that used in the grafting procedure, and that derived from the grafted pyrimidine derivative) and creation of sulfur-based materials from the heterocyclic derivatives leads to a change in the acid base characterization. Figure 5 shows the pHpzc results of both sorbents (MCH and MC-TDP) by investigation of the pH (drift method). The pH_PZC_ was apparently changed (from 6.059 to 5.66) with the modification from MCH to MC-TDP, respectively, by successive addition of the TDP substrate. This illustrated the completed deprotonation of the sorbent after these values smarten the electron pairs for chelation to the positively charged metal ions (or complexes) in which the functional groups were bearing a negative charge.

### 3.2. Sorption Properties in Synthetic Solutions

#### 3.2.1. pH Effect

Appendix A shows the distribution of chromium species under the selected experimental conditions. The pH values have a significant effects on the dissociation of active groups, which exists at the sorbent surface that is used for metal binding; on the other hand, the pH also affects the metal speciation. At acidic values of pH, CrO_3_SO_4_^2^, Cr_2_O_7_^2−^ and HCrO_4_^−^ are the most common anionic chromium species that are found in the solution, while at pH > 5, CrO_4_^2−^ appeared and predominates at pH > 6.46.

Figure 6a compares the sorption capacities of both duplicated experiments (for reproducibility) as a function of pH equilibrium (pH_eq_) for both MC-TDP (functionalized chitosan NP) and MCH (reference crosslinked chitosan particles).

From the identification of sorption values of both experiments, it was concluded there was a successive reproducibility of the sorbents. Enhancing sorption as pH increased (maximum values at pH 3–4) is due to increasing the negatively charged chromate complex species with the positively charged functional groups on the sorbent surface. The increase in sorption performance as functionalization was performed, compared to the non-functionalized composite, shows the valorization of the grafting procedure. On the other hand, the sorption capacity raised from 0.9 to 3.34 mmol Cr g^−1^ for crosslinked chitosan (MCH) and modified sorbent (MC-TDP), respectively, at pH 4 and pH 3 for the two sorbents, respectively; this can be an assisting factor for the recovery of chromate ions in the acidic medium [68,69]. Vieira et al. [70] found efficient reduction results of the chromate ions on various types of membrane of chitosan. Figure 6b shows the pH variation to be higher by around 0.4 and lower by around 0.3 for the MCH (reference material) and MC-TDP (functionalized sorbent), respectively. Figure 6c reports the results of plotting (log10 D units) and pH_eq_. The results curves are split into two different sections. The determined slope for each section shows the curves with acidic medium +0.35 for MCH and +0.62 for MC-TDP. The slope data were calculated (for the right side) to be close to −0.13 for MCH and −0.35 MC-TDP, which indicates the use of ion exchange in the sorption procedure during Cr adsorption.

#### 3.2.2. Uptake Kinetics

Figure 7 shows the sorption kinetics of both MCH and MC-TDP sorbents at different time periods (pH_in_ 3). It shows that the maximum equilibrium was achieved during the first 60 min and 35 min for the MCH (the reference material) and the functionalized sorbent (MC-TDP), respectively; this time is sufficient for complete saturation, which reflects the fast sorption kinetics of the modified sorbent compared with the reference material, and it reflects the effect of the modification process on improving the kinetic properties as well as the sorption capacity (as discussed in the pH section). The sorbent was synthesized in nanoscale size with a thin organic layer on the magnetite, which limits the intraparticle diffusion for chromate ions in the pores of the sorbent (not highly porous; see the BET data); this is the reason for the fast sorption kinetics. The models were listed with details in Appendix A, while the equilibrium parameters and capacities (calculated and experimental values) were reported in Table 2. Regardless of the sorption equilibrium (experimental vs. calculated), it confirms that the PFORE (pseudo-first order rate equation) is the most fitted model compared to the PSORE (the pseudo-second order rate equation) and RIDE (resistance to the intraparticle diffusion).

Figure 7 shows the MCH sorption kinetics and the two repeated experiments for MC-TDP, while Appendix A shows the unfitted models (PSORE and RIDE). Interesting interpretations were discussed by Hubbe [71] and also Simonin [72]. The type of mechanism (physical vs. chemical sorption) should depend on the experimental condition, which controls this type of interaction. As expected from the study, the Cr removal was greater via MC-TDP sorbent than via MCH sorbent.

#### 3.2.3. Sorption Isotherms

Sorption isotherms are used for evaluation of the process of the sorbent affinities toward metal ion uptake (the residual concentration of chromate) as well as calculating the maximum sorption capacities for each sorbent (MCH and MC-TDP). The parameters which describe the sorption isotherms models are shown in Appendix A. The modified grafted sorbent shows a good steep sorption profile before tendency to saturation plateau. The modified functionalized sorbent (MC-TDP) and the crosslinked chitosan (MCH; the reference material) show a remarkable reproducibility by fitting both sets of experimental data, as represented in Figure 8 of the two repeated experiments. The sorption properties of MC-TDP were compared with the crude material (MCH). The sorption performance is determined, in which it was increased to the maximum and the initial slope is according to MC-TDP > MCH, which follows the same order of the q_m_ as reported in Table 3. The models were fitted by comparing the sorption capacities of the fitted models and the experimental one for evaluation of the sorption isotherms (Langmuir, Freundlich and Sips equations, as reported in Appendix A).

The Langmuir and Sips models were the most fitting of the sorption isotherms, compared to the unfit Freundlich equation (Appendix A). It is noteworthy that the Langmuir equation corresponds to a mechanistic model; in other words, it performed without interaction between sorbent molecules. The Freundlich equation is considered a mathematical equation with the heterogeneous sorption, which has a hypothesis with regard to the possible interactions of the sorbed molecules. The other model (Sips equation) shows the combination between the Freundlich and Langmuir, with the parameter (n) making it easier to fit the data (experimental data).

#### 3.2.4. Interaction Mechanism

From the sorption results (mainly the pH effect, as an example) and the sorbent properties (i.e., the pHpzc, and FTIR analysis) as well as the speciation diagram of chromate ions, it was indicated that the sorbent is completely protonated (check the pHpzc properties) during the optimum sorption properties (pH 4 for the MC-TDP). The interaction was expected to be carried out with the protonated NH and OH (decrease in the intensities of these functional groups as Cr sorption was performed; from the FTIR) and the negatively charged chromate anions. The FTIR also showed the new bands of the protonated ammonium species (1380 cm^−1^), which emphasized the binding possibility by electrostatic attraction. Appendix A shows the binding route of the functionalized chitosan and the chromate ions in the acidic medium, which shows the participation of OH and NH on the sorption mechanism through electrostatic attraction of positively charged functional groups and negatively charged metal species. Equations (1)–(4) show the expected reduction of chromium VI to III for facilitating the sorption properties.
-C-H _(org)_ + Cr(VI) + H_2_O_(aq)_ = -C-OH_(org)_ + Cr(III) + 2H^+^(1)
-C=O_(org)_ + Cr(VI) + H_2_O_(aq)_ = -COOH_(org)_ + Cr (III) + H^+^(2)
Cr_2_O_7_^2−^ + 6Fe^2+^(s) + H_2_O_(aq)_ + 6H^+^ = 2Cr^3+^ + 8OH^−^ + 6Fe^3+^(s)(3)
Surface C ^(+)^ + Cr_2_O_7_^2−^ = Surface C-Cr_2_O_7_ (Surface)(4)
where (s) means solid.

#### 3.2.5. Metal Desorption and Sorbent Recycling

Metal ions desorption and sorbent recycling are the most important parameters for the sorption evaluation process. The recycling process was performed for the both sorbents (MCH and MC-TDP). From previous work [73,74,75], there is limited degradation of magnetic core components (around 1.5%) after 5 cycles by using around 0.5 M HCl solution; herein, 0.2 M HCl is sufficient for complete desorption, so the loss is negligible. Table 4 reports the five cycles of sorption/desorption on the sorbent loading performances. It represents a limited loss in the sorption efficiency: i.e., 3.1 and 2.78% for MCH and MC-TDP, respectively. Thus, the high performance stability of the sorbent was noticeable; this is indicated in the FTIR spectra (Figure 1), which illustrated restoration of the functional groups after being used in the binding to chromate.

Appendix A compares desorption kinetics of the sorbents collected from the uptake kinetic experiments. It shows the fast desorption process with high performance of desorbing reagent/0.2 M HCl. The complete desorption was reached within 20 min, which is fast compared to the loading processes, thus indicating other advantages of using such sorbents for heavy metal removal and water decontamination.

### 3.3. Treatment of Tannery Wastewater

The tannery effluents contain a high percent of chromium associated with other elements with the following concentrations: 215.31, 0.117, 0.334, 0.513 and 0.488 mmol M L^−1^ of Cr(VI), Pb(II), Cd(II), Ni(II) and Co(II), respectively. The chromate ions are in large excess, from 644 times for Cd(II), 1840 times for Pb(II) and around 430 times (±10) for both Co(II) and Ni(II). It is noteworthy that the level of chromate concentration is considered to be too high to create a competitive process for sorption (usually, to make sense of the sorption, the concentration should usually be below around 150 (±50) mg L^−1^). The chromate is a high concentration to be consumed by the normal sorbent dose (SD: g L^−1^) as used in the synthetic solutions (around 0.6 g L^−1^). The high SD was used for efficient removal of chromate ions (around 25 g L^−1^). Evaluation of the sorption properties was performed by comparing the removal efficiency of grafted sorbent at pH 4. The SC (selectivity coefficients) and the R% (recovering efficiency) are represented in Figure 9 at different SD values. Appendix A shows the efficient removal of MC-TDP sorbent for the chromium ions.

As expected, as the pH values increased, a high removal efficiency was produced, especially for the modified grafted sorbent. This is due to the protonation of the sorbents at the acidic pH values, which causes a repulsion with chromium (VI) ions (source as sulfate species, which yield a positively charged chromium species (as cations)).

The removal reached 99.57% for MC-TDP at pH 4. In terms of selectivity, it was found that the MC-TDP is around 35–152, and the selectivity (SC) has the order Cd(II), Pb(II), Co(II), and Ni(II), while the recovery has the order Cr(VI), Cd(II), Pb(II), Ni(II), and Co(II).

As expected, the higher SD values of the sorbent with the high complex polymetallic solution made it difficult to reach the standard values for the drinking water, but there was a strong fit with regard to the irrigation and the livestock drinking water levels.

## 4. Conclusions

This research article shows an itemized overview for biosorption of chromium in a multi-step study, providing in detail the sorption efficiency of Cr ions. The modified chitosan sorbent grafted with synthesized heterocyclic pyrimidine derivative from cyclization reaction of diethyl malonate and thiourea was successfully grafted on activated chitosan to produce MC-TDP sorbent. The synthesized sorbent was investigated by mass spectroscopy, FTIR, BET (surface area), TGA, EA, SEM and SEM-EDX analyses. The effect of sorption parameters (pH effect, uptake kinetics and sorption isotherms) was investigated, and the removal efficiency toward Cr(VI) were evaluated through application on synthetic and nature tannery effluent solution. The biosorption kinetics and the related models of equilibrium were also discussed. The sorption kinetics of MC-TDP is around 45 min with the PFORE fitted model, in which the Langmuir and Sips equations fit the sorption isotherms. The maximum sorption for the MC-TDP is close to 5.7 mmol Cr(VI) g^−1^ compared to 1.97 mmol Cr(VI) g^−1^ for the crosslinked chitosan MCH (as reference material, sorbent without functionalization). This sorbent was tested for Cr recovery from a polymetallic contaminated tannery effluent solution. The results showed its good efficiency in the removal process by application of high sorbent doses (SD; around 25–30 g L^−1^). The sorbent shows promising results for the decontamination of waste water. Overall, researchers can use this article to gain fundamental and valuable knowledge, which in turn will be helpful in their planned related research for Cr removal or water treatment technology.

## Figures and Tables

**Figure 1 materials-15-04676-f001:**
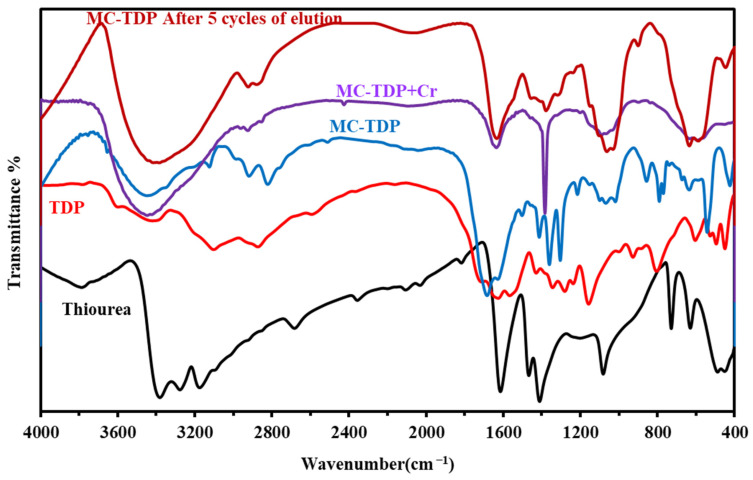
FTIR of thiourea, TDP, functionalized chitosan with TDP (MC-TDP), MC-TDP + Cr and after five cycles of sorption–desorption.

**Figure 2 materials-15-04676-f002:**
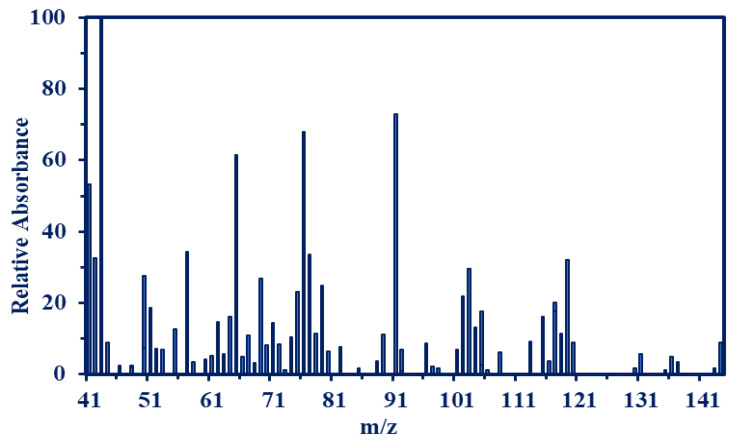
Mass spectroscopy for the synthesized TDP moiety.

**Figure 3 materials-15-04676-f003:**
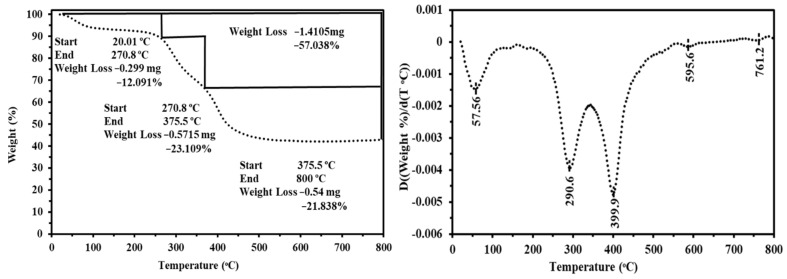
TGA and the DrTGA thermal analyses of modified sorbent (MC-TDP).

**Figure 4 materials-15-04676-f004:**
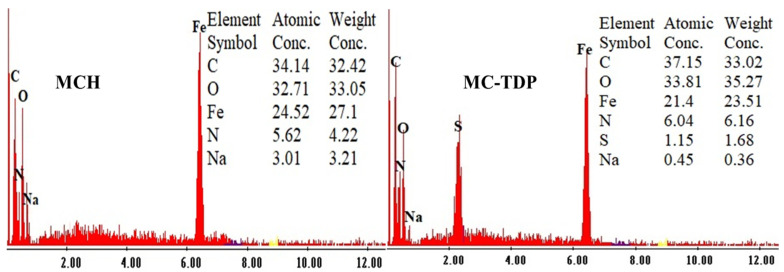
EDX analysis of MCH and MC-TDP.

**Figure 5 materials-15-04676-f005:**
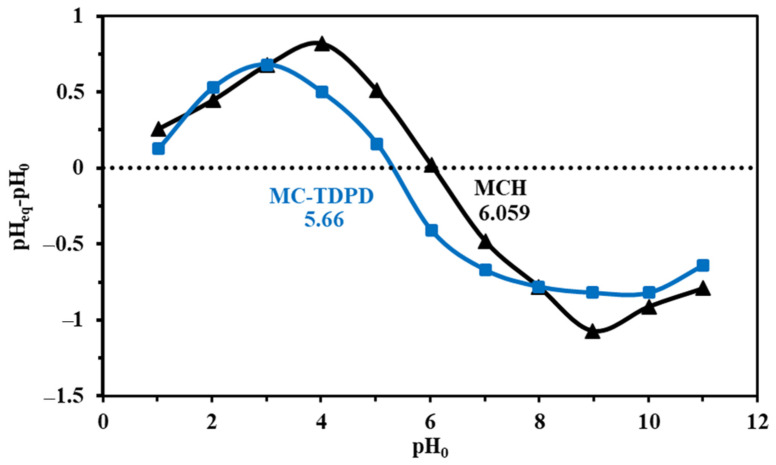
pHpzc of non-modified MCH and the final modified sorbent MC-TDP (pH range 1–14).

**Figure 6 materials-15-04676-f006:**
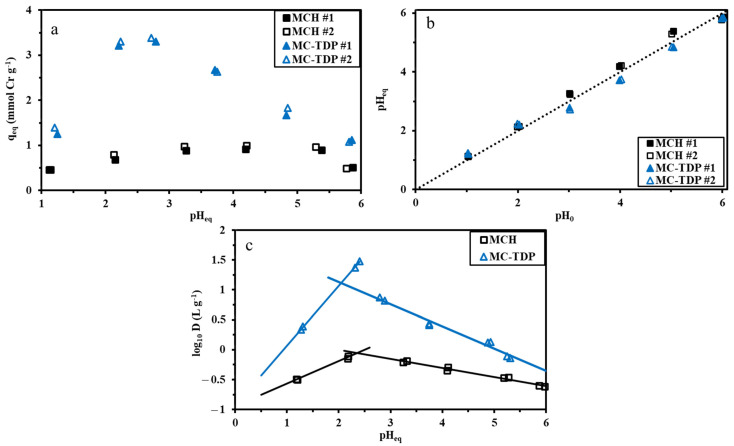
The pH study of MCH and MC-TDP on adsorption performance for chromate ions (**a**), pH variation (**b**), and results of log_10_ D (**c**) against pH_eq._

**Figure 7 materials-15-04676-f007:**
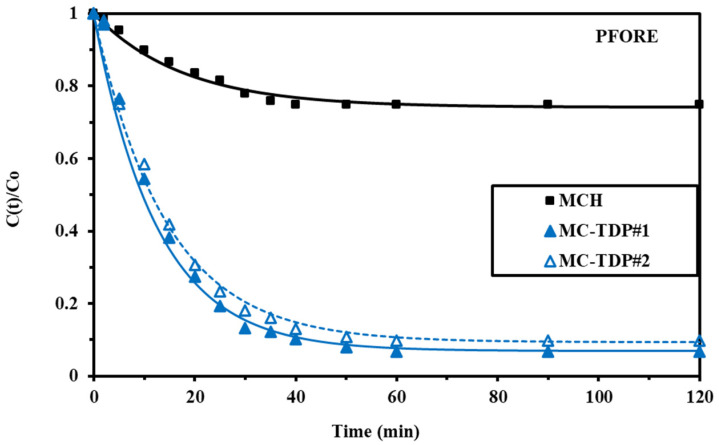
Uptake Cr(VI) kinetics on MCH, and MC-TDP modeling with the PFORE.

**Figure 8 materials-15-04676-f008:**
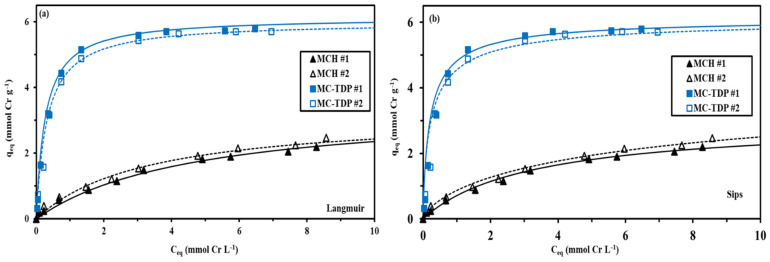
Sorption isotherms of Cr(VI) using MCH, MC-TDP; Langmuir (**a**) and Sips (**b**) modeling equations.

**Figure 9 materials-15-04676-f009:**
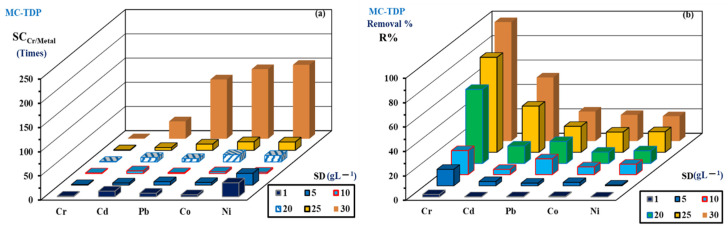
SC (selectivity coefficient) (**a**) and R% (removal %) (**b**) of chromium ions during treatment of tannery effluents by MC-TDP.

**Table 1 materials-15-04676-t001:** Elemental analysis of MCH and MC-TDP sorbents.

Sorbent	C	N	H	O	S
	[%]	[%]	[%]	[%]	[%]
MCH	34.65	3.98	4.97	27.14	
MC-TDPD	37.09	5.63	5.15	32.96	1.11

**Table 2 materials-15-04676-t002:** Parameters of the PFORE, PSORE and RIDE for chromium(VI) sorption kinetics for the experiments.

		MCH	MC-TDP	
Model	Parameter	1	1	2
Exp.	q_eq_ (mmol g^−1^)	0.97	3.32	3.19
PFORE	q_eq,1_ (mmol Cr g^−1^)	1.01	3.28	3.21
	k_1_ × 10^2^ (min^−1^)	4.98	3.28	2.77
	R^2^	0.997	0.989	0.994
	AIC	−74.5	−126	−117
PSORE	q_eq,2_ (mmol Cr g^−1^)	1.39	3.51	3.27
	k_2_ × 10^2^ (L mmol^−1^ min^−1^)	2.87	2.86	3.22
	R^2^	0.863	0.768	0.733
	AIC	−15.4	−26.5	−47.3
RIDE	D_e_ × 10^8^ (m^2^ min^−1^)	9.97	5.76	6.48
	R^2^	0.925	0.89	0.76
	AIC	−42.1	−56.4	−61.4

**Table 3 materials-15-04676-t003:** Modeling of Cr(VI) sorption isotherms for the MCH and MC-TDP sorbents.

			MCH		MC-TDP	
Model	Parameter		1	2	1	2
Experimental	q_m,exp._	mmol g^−1^	1.97	2.01	5.76	5.69
Langmuir	q_m,L_	mmol g^−1^	2.01	1.97	5.81	5.61
	b_L_	L mmol^−1^	0.967	0.874	2.18	2.97
	R^2^	-	0.9846	0.9931	0.9937	0.9894
	AIC	-	−65.88	−84.38	−127.27	−132.28
Freundlich	k_F_	mmol^1−1/n^g^−1^ L^1/n^	0.896	0.906	2.18	1.86
	n_F_	-	1.86	2.185	2.18	3.53
	R^2^	-	0.8954	0.7998	0.7968	0.8164
	AIC	-	−25.53	−31.91	−36.182	−41.37
Sips	q_m,S_	mmol g^−1^	1.23	1.98	5.697	5.721
	b_S_	L mmol^−1^	0.765	0.8876	1.12	1.434
	n_S_	-	1.32	1.48	2.17	2.65
	R^2^	-	0.9786	0.9946	0.9956	0.9895
	AIC	-	−123.63	−163.94	−97.68	−117.68

**Table 4 materials-15-04676-t004:** Sorption/desorption recycling of MCH and MC-TDP.

Cycles	MCH				MC-TDP			
	Removal (%)	S.D. (R.%)	Desorption (%)	S.D. (De%)	Removal (%)	S.D. (R. %)	Desorption (%)	S.D. (De%)
1	24.18	0.21	100	0.2	78.19	0.85	99.14	0.95
2	23.54	0.98	99.31	0.73	77.49	0.55	99.65	0.5
3	22.69	0.94	100	0.61	76.86	0.04	99.19	1.17
4	22.51	0.89	100	0.49	76.65	0.52	99.65	0.36
5	21.7	1.13	99.92	0.97	75.59	0.87	99.44	0.56

## Data Availability

Data are available from authors.

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
