# Peer review of "Synthesis and Characterization of Functionalized Chitosan Nanoparticles with Pyrimidine Derivative for Enhancing Ion Sorption and Application for Removal of Contaminants"

_materials, 2022, doi:10.3390/ma15134676_

Round 1

Reviewer 1 Report

This ms reports the functionalization of chitosan with 2-thioxodihydropyrimidine-4,6(1H,5H)-dione for the removal of hexavalent chromium. The obtained NPs were characterized by using different techniques. The sorption of Cr(VI) by CHI-based NPs was characterized by performing isotherms and kinetics experiments. The ms is well written and the data sounds. The discussion could be improved. For example, it is understandable the interaction between Cr(VI) as anionic species and protonated ammonium chitosan but the same does not happen with other metal ions.

References 7 and 8 are not appropriate as papers describing the properties of heavy metals and their presence in water. For that authors must cite other papers such as 10.1016/j.jenvman.2019.05.126; 10.1016/j.eti.2021.101504.

The name of reagents (e.g., Lines 74 and 76) are not proper names….they must be written with lower case.

DA, PFORE, PSORE, RIDE, etc. must be defined (Line 94)

The symbol for chitosan should be other than “C”; for example “CHI” or “CS”.

Figure 1 is quite crowded and must be simplified or, at least, the wavenumbers must be skipped from the figure.

The quality of Figure 2 should be improved.

The maximum degradation temperatures at 596 ºC and 761 ºC are quite speculative and need further analysis for a correct evaluation (for example, by changing the heating rate); in fact, authors do not give any explanation for the occurrence of such degradation steps.

Figures 5 should be better visualized by using bars instead of data points.

The number of decimal figures for removal and S.D. in table 4 must be decreased.

Author Response

Response to Reviewers

Red: Response to the comments of Reviewers

Blue: New information added to the manuscript

We would like to thank the Reviewer for his instructive and meaningful comments. We try to modify the work probably more appropriate for the whole manuscript. We hope the Reviewer and the Editor will accept the modification and corrections that performed on the new version.

Open Review

English language and style

( ) Extensive editing of English language and style required
( ) Moderate English changes required
(x) English language and style are fine/minor spell check required
( ) I don't feel qualified to judge about the English language and style

Yes

Can be improved

Must be improved

Not applicable

Does the introduction provide sufficient background and include all relevant references?

( )

(x)

( )

( )

Are all the cited references relevant to the research?

( )

(x)

( )

( )

Is the research design appropriate?

(x)

( )

( )

( )

Are the methods adequately described?

(x)

( )

( )

( )

Are the results clearly presented?

(x)

( )

( )

( )

Are the conclusions supported by the results?

( )

(x)

( )

( )

Comments and Suggestions for Authors

 This ms reports the functionalization of chitosan with 2-thioxodihydropyrimidine-4,6(1H,5H)-dione for the removal of hexavalent chromium. The obtained NPs were characterized by using different techniques. The sorption of Cr(VI) by CHI-based NPs was characterized by performing isotherms and kinetics experiments. The ms is well written and the data sounds. The discussion could be improved. For example, it is understandable the interaction between Cr(VI) as anionic species and protonated ammonium chitosan but the same does not happen with other metal ions.

 Thanks for the positive evaluation of our work. We add scheme of interaction and discuss the probability of reduction of Cr(VI)  to Cr(III)

This part was added

3.2.4. Interaction Mechanism

From the sorption data (mainly the pH effect) and the sorbent characterization from the pHpzc, and FTIR analysis as well as the speciation diagram of chromate ions. It was indicated that the sorbent is completely protonated (check the pHpzc properties) during the optimum sorption properties (pH 4 for the MC-TDP). The interaction was performed with the protonated NH and OH (decreased the intensities of these functional groups as Cr sorption was performed; from the FTIR) and the negatively charged chromate anions. The FTIR also appeared the new bands of N+ for the protonated ammonium species. Scheme S4 shows the binding route of the functionalized chitosan and the chromate ions in the acidic medium, which shows the participation of OH and NH on the sorption mechanism through electrostatic attraction of positively charged functional groups and negatively charged metal species. Equations 1-4 show the expected reduction of chromium VI to III for facilitating the sorption properties.

-C-H + Cr(VI) + H2O  =  -C-OH + Cr(III) + H+ ……(Eq.1)

-C=O + Cr(VI) + H2O  =  -COOH + Cr (III) + H+ …….( Eq.2)

Cr2O72- + Fe2+(s) + H2O  =  Cr3+ + OH- + 6Fe3+(s) …….( Eq.3)

Surface C (+) + Cr2O72- = Surface C - Cr2O7 (Surface)…….( Eq.4)

And this was added in the SI

Scheme S4. Expected binding mechanism of chromate ions with MC-TDP sorbent at acidic pH medium

References 7 and 8 are not appropriate as papers describing the properties of heavy metals and their presence in water. For that authors must cite other papers such as 10.1016/j.jenvman.2019.05.126; 10.1016/j.eti.2021.101504.

Thanks for notification, we add the suggested references and delete the others

The name of reagents (e.g., Lines 74 and 76) are not proper names….they must be written with lower case.

Thanks, it was changed in the whole manuscript

DA, PFORE, PSORE, RIDE, etc. must be defined (Line 94)

Thanks for your comment, we defined the DA and the PFORE (L36) while the PSORE and RIDE are defined in L335 and L336

The symbol for chitosan should be other than “C”; for example “CHI” or “CS”.

Thanks for alerting, we usually use the magnetite chitosan as MCH or MC, and it was corrected to CHI

Figure 1 is quite crowded and must be simplified or, at least, the wavenumbers must be skipped from the figure.

Thanks for alerting, it was changed by removing the wavenumbers

The quality of Figure 2 should be improved.

Thanks, it was improved, hope it will meet your satisfactions, and the original figure was withdrawn to the SI.

The maximum degradation temperatures at 596 ºC and 761 ºC are quite speculative and need further analysis for a correct evaluation (for example, by changing the heating rate); in fact, authors do not give any explanation for the occurrence of such degradation steps.

Actually this is the usual analysis that was performed for measuring the thermal degradation of the organic materials, and to support the idea of the reviewer, we burn the sample to 900 ºC and measuring the total loss (%), it was found to be 58.4%, that is closed to the results that obtained by the TGA analysis (57.038).  

This paragraph was added

On the other hand, the sorbent was subjected to burn in vacuum oven at 900 ºC for two hours and the total loss is around 58.1 which is closed to that obtained from the TGA analysis.

Figures 5 should be better visualized by using bars instead of data points.

Thanks for alerting, we changed it.

The number of decimal figures for removal and S.D. in table 4 must be decreased.

Thanks, it was modified

Cycles

MCH

MC-TDP

Removal (%)

S.D. (R.%)

Desorption (%)

S.D. (De%)

Removal (%)

S.D. (R. %)

Desorption (%)

S.D. (De%)

1

24.18

0.21

100

0.2

78.19

0.85

99.14

0.95

2

23.54

0.98

99.31

0.73

77.49

0.55

99.65

0.5

3

22.69

0.94

100

0.61

76.86

0.04

99.19

1.17

4

22.51

0.89

100

0.49

76.65

0.52

99.65

0.36

5

21.697

1.1275

99.923

0.968

75.589

0.868

99.442

0.563

Reviewer 2 Report

This study describes the synthesis and Characterization of Functionalized Chitosan Nanoparticles with Pyrimidine derivatives for Enhancing Ion Sorption and Application for Removing Contaminants. Several drawbacks and academic inconsistencies limit the acceptance of this work. Please consider the following observations:

  1. ENGLISH GRAMMAR. The manuscript must be thoroughly revised by an English native speaker.
  2. The authors discuss the use of modified chitosan for the removal of metal ions from aqueous effluents. For a practical application, it is not a cost-effective approach (the synthesis scheme of this work is expensive). There are other polymers or adsorbents of lower cost and high efficiencies for metal ion removal (a) Sun, Gang, and Weixing Shi. "Sunflower stalks as adsorbents for the removal of metal ions from wastewater." Industrial & engineering chemistry research 37, no. 4 (1998): 1324-1328.; b) Qasem, Naef AA, Ramy H. Mohammed, and Dahiru U. Lawal. "Removal of heavy metal ions from wastewater: A comprehensive and critical review." NJ Clean Water 4, no. 1 (2021): 1-15.)
  3. Page 3, lines 115-118: "The second step of the synthesis is the preparation of chitosan magnetite nanoparticles through dissolving 4 g chitosan in 100 mL 7% acetic acid solution, the addition of 4 g of the prepared magnetite powder with continuous stirring. The mixture was precipitated by adjusting the pH to 10 using NaOH solution." It is difficult to visualize the procedure just described. Magnetite dissolves in acidic media, thus the authors do not incorporate such material in the polymer. Maybe ion insertion and different iron ion-containing chemical species. The authors must include an image (photographs of the different materials under study).
  4.  The authors report on using potassium dichromate as a chromate source. Figure S6 shows a black solution labeled as chromate solution before treatment. The solution should be orange!
  5.  

Author Response

Response to Reviewer#1

Red: Response to the comments of Reviewers

Blue: New information added to the manuscript

First, we would like to thank the Reviewers for their time consumed in the deep reviewing of the paper and for their instructive and meaningful comments. We try to modify this work probably more appropriate for the whole manuscript. We hope the Reviewer and the Editor will accept the modification and corrections that performed on the new version.

Open Review

English language and style

(x) Extensive editing of English language and style required
( ) Moderate English changes required
( ) English language and style are fine/minor spell check required
( ) I don't feel qualified to judge about the English language and style

Yes

Can be improved

Must be improved

Not applicable

Does the introduction provide sufficient background and include all relevant references?

( )

( )

(x)

( )

Are all the cited references relevant to the research?

( )

( )

(x)

( )

Is the research design appropriate?

( )

(x)

( )

( )

Are the methods adequately described?

( )

( )

(x)

( )

Are the results clearly presented?

( )

(x)

( )

( )

Are the conclusions supported by the results?

( )

(x)

( )

( )

Comments and Suggestions for Authors

This study describes the synthesis and Characterization of Functionalized Chitosan Nanoparticles with Pyrimidine derivatives for Enhancing Ion Sorption and Application for Removing Contaminants. Several drawbacks and academic inconsistencies limit the acceptance of this work. Please consider the following observations:

Thanks for your interpretation and recommendation. We considered with attention this suggestion;

  1. ENGLISH GRAMMAR. The manuscript must be thoroughly revised by an English native speaker.

Thanks for your comment; we revised the whole manuscript and hope it is fine now

  1. The authors discuss the use of modified chitosan for the removal of metal ions from aqueous effluents. For a practical application, it is not a cost-effective approach (the synthesis scheme of this work is expensive). There are other polymers or adsorbents of lower cost and high efficiencies for metal ion removal (a) Sun, Gang, and Weixing Shi. "Sunflower stalks as adsorbents for the removal of metal ions from wastewater." Industrial & engineering chemistry research 37, no. 4 (1998): 1324-1328.; b) Qasem, Naef AA, Ramy H. Mohammed, and Dahiru U. Lawal. "Removal of heavy metal ions from wastewater: A comprehensive and critical review." NJ Clean Water 4, no. 1 (2021): 1-15.)

Thanks for alerting; Actually, the work is based on the biopolymer (safe to the environment and safe at the end-of-life cycles) the main target of this work is to synthesis a highly removal Cr and efficient apply to the highly contaminated solution with improving the kinetic efficiency. The Sun, Gang, and Weixing Shi work have capacity toward Cr of around 25.07 mgg-1 comparing to 301 mgg-1 of our work and this is the most advantage as well as the reuses and stability.

  1. Page 3, lines 115-118: "The second step of the synthesis is the preparation of chitosan magnetite nanoparticles through dissolving 4 g chitosan in 100 mL 7% acetic acid solution, the addition of 4 g of the prepared magnetite powder with continuous stirring. The mixture was precipitated by adjusting the pH to 10 using NaOH solution." It is difficult to visualize the procedure just described. Magnetite dissolves in acidic media, thus the authors do not incorporate such material in the polymer. Maybe ion insertion and different iron ion-containing chemical species. The authors must include an image (photographs of the different materials under study).

The synthesis parameters condition is a conclusion from several work that studies the synthesis and modification of magnetite chitosan nano/micro particles[1-7]. Another studies were performed on the degradation percent of magnetite using acidic solution of 0.5 M HCl and a limited loss was noticed after 5 cycles of sorption desorption (less than 1% using 0.5M HCl)after 5 cycles of sorption desorption [7, 8].  We add the below figure to indicate the efficient separation of magnetite chitosan by external magnetic bar.

[1] M.F. Hamza, F.Y. Ahmed, I. El-Aassy, A. Fouda, E. Guibal, Groundwater purification in a polymetallic mining area (SW Sinai, Egypt) using functionalized magnetic chitosan particles, Water, Air, & Soil Pollution 229(11) (2018) 1-14.

[2] M.F. Hamza, M.M. Aly, A.A.-H. Abdel-Rahman, S. Ramadan, H. Raslan, S. Wang, T. Vincent, E. Guibal, Functionalization of magnetic chitosan particles for the sorption of U (VI), Cu (II) and Zn (II)—Hydrazide derivative of glycine-grafted chitosan, Materials 10(5) (2017) 539.

[3] M.F. Hamza, A. Fouda, K.Z. Elwakeel, Y. Wei, E. Guibal, N.A. Hamad, Phosphorylation of guar gum/magnetite/chitosan nanocomposites for uranium (VI) sorption and antibacterial applications, Molecules 26(7) (2021) 1920.

[4] M.F. Hamza, A. Gamal, G. Hussein, M.S. Nagar, A.A.H. Abdel‐Rahman, Y. Wei, E. Guibal, Uranium (VI) and zirconium (IV) sorption on magnetic chitosan derivatives–effect of different functional groups on separation properties, Journal of Chemical Technology & Biotechnology 94(12) (2019) 3866-3882.

[5] M.F. Hamza, A.E.-S. Goda, S. Ning, H.I. Mira, A.A.-H. Abdel-Rahman, Y. Wei, T. Fujita, H.H. Amer, S.H. Alotaibi, A. Fouda, Photocatalytic Efficacy of Heterocyclic Base Grafted Chitosan Magnetite Nanoparticles on Sorption of Pb (II); Application on Mining Effluent, Catalysts 12(3) (2022) 330.

[6] M.F. Hamza, Y. Wei, A. Benettayeb, X. Wang, E. Guibal, Efficient removal of uranium, cadmium and mercury from aqueous solutions using grafted hydrazide-micro-magnetite chitosan derivative, Journal of Materials Science 55(10) (2020) 4193-4212.

[7] M.F. Hamza, Y. Wei, H. Mira, A.-H. Adel, E. Guibal, Synthesis and adsorption characteristics of grafted hydrazinyl amine magnetite-chitosan for Ni (II) and Pb (II) recovery, Chemical Engineering Journal 362 (2019) 310-324.

[8] A. Benettayeb, A. Morsli, K.Z. Elwakeel, M.F. Hamza, E. Guibal, Recovery of heavy metal ions using magnetic glycine-modified chitosan—application to aqueous solutions and tailing leachate, Applied Sciences 11(18) (2021) 8377.

The below photo concerning with collection of magnetite chitosan particles from solution which indicate the efficient of incorporation the magnetite in the organic material

  1. The authors report on using potassium dichromate as a chromate source. Figure S6 shows a black solution labeled as chromate solution before treatment. The solution should be orange!

You is right but this solution is a dark green solution produced after the treatment from tannery industries and considered as waste solution (tannery effluents) that enriched with other metal ions as Cd, Co, Ni and Pb beside Cr ions beside organic materials

Reviewer 3 Report

1. Mention in the introduction the currently applied, most popular methods of Cr(VI) removal (except those based on polysaccharides). What benefits bring the modification/exchange of these methods with materials being an object of this study?

2. Please indicate clearly why modification of chitosan devoted to sorption processes is needed. The introduction: the "second" part dedicated to chitosan needs explanation.

3. The introduction should provide the knowledge/background for the aim of this work. In its current form, it is not enough. How and why can be the proposed materials better in Cr sorption than others? What problem will be solved/benefits? And finally – the novelty should be clearly seen. The main question is why such a modification has been proposed? Why were magnetic particles prepared?

4. In the whole manuscript, the results are given and described; however, there is a lack of any discussion – on why these changes occur, how these changes influence the material properties (are these changes positive or negative), and how they can affect the sorption properties of the material

5. Conclusions should also be improved. Why does the modification improve the sorption properties?

Line 25: explain all abbreviations when they are used for the first time (e.g., FTIR, BET, etc.)

Line 25-29: sentence need revision (style)

Line 29: what kind of conditions?

Line 29: "high sorption uptake" – high Cr(VI) sorption?

Line 41” "suchlike" – should be "such like"

Line 54: what is the limit?

Line 62: not "absorbtion" but "adsorption – also check the other parts of the manuscript

Line 70 – to general sentence. What does "stability" means?

Line 95: as the Authors stated that DA of chitosan is an essential parameter for its properties, the exact value of DA should be determined

Line 130: 'Mg-chitosan" suggest not magnetic but chitosan with Mg element – consider other abbreviation

FTIR analysis; please shortly describe, based on the observed changes, what are the main results regarding the sorption and desorption process,

Line 227: not possible that the evaporation of water occurs at 270.8 C.deg (see DTG), this is the end of evaporation, however, max should be given (or better, onset temperature)

Line 264: Check the chemical formulas

Line 348: are the numbers in % or in the difference in sorption (indicate the unit)

Section 2.3 – explain SD

Figure 8: explain axis notation (SC, R%, SD)

Check all parts f the manuscript: be sure to write "Cr(VI)" not "Cr (VI)".

English: "reaction maintained stirring" (line 121), "separated" (instead of "separation", line 125) etc

Mirror mistakes: line 90 (N2), line 184 (cm-1), line 190 (pikes), line 237 (DrTG), line 232 (is closed to) etc. – please carefully read the whole manuscript.

There are two schemes S1 in supplementary materials; see TDPD in Table S2 (lack of TDPD explanation, what is the difference between TDPD and TDP).

Author Response

First, we would like to thank the Reviewers for their general appreciation of the work and for their instructive and meaningful comments.

Response to Reviewer#2

Red: Response to the comments of Reviewers

Blue: New information added to the manuscript

Open Review

English language and style

( ) Extensive editing of English language and style required
(x) Moderate English changes required
( ) English language and style are fine/minor spell check required
( ) I don't feel qualified to judge about the English language and style

Yes

Can be improved

Must be improved

Not applicable

Does the introduction provide sufficient background and include all relevant references?

( )

( )

(x)

( )

Are all the cited references relevant to the research?

( )

(x)

( )

( )

Is the research design appropriate?

( )

(x)

( )

( )

Are the methods adequately described?

(x)

( )

( )

( )

Are the results clearly presented?

(x)

( )

( )

( )

Are the conclusions supported by the results?

( )

( )

(x)

( )

Comments and Suggestions for Authors

  1. Mention in the introduction the currently applied, most popular methods of Cr(VI) removal (except those based on polysaccharides). What benefits bring the modification/exchange of these methods with materials being an object of this study?

Thanks for your comment, it was revised well and supported by further methods for Cr(VI) removal.

This part was added

The treatment of such contaminated water/ effluents is a critical point in these industries, to minimize their impact toward water bodies and environment [18]. Various techniques were be used for this mission including electrochemical reduction [19], precipitation [20], (after reduction of Cr(VI) to Cr(III) [21]), or bio-reduction [22]. The acidic medium enhance reduction of Cr(VI) to Cr(III) for easily precipitation. Solvent extraction [23,24] and the impregnated process (solvent-impregnated resins) are also used for the recovering and of chromate anions [25-27]. Other studies based on using of zero-valent iron, iron oxide, and magnetite sorbents [28,29] for the removal of chromate anions.

Also we add a mechanism for binding to be more understandable for the reader

3.2.4. Interaction Mechanism

From the sorption data (mainly the pH effect) and the sorbent characterization from the pHpzc, and FTIR analysis as well as the speciation diagram of chromate ions. It was indicated that the sorbent is completely protonated (check the pHpzc properties) during the optimum sorption properties (pH 4 for the MC-TDP). The interaction was performed with the protonated NH and OH (decreased the intensities of these functional groups as Cr sorption was performed; from the FTIR) and the negatively charged chromate anions. The FTIR also appeared the new bands of N+ for the protonated ammonium species. Scheme S4 shows the binding route of the functionalized chitosan and the chromate ions in the acidic medium, which shows the participation of OH and NH on the sorption mechanism through electrostatic attraction of positively charged functional groups and negatively charged metal species. Equations 1-4 show the expected reduction of chromium VI to III for facilitating the sorption properties.

-C-H + Cr(VI) + H2O  =  -C-OH + Cr(III) + H+ ……(Eq.1)

-C=O + Cr(VI) + H2O  =  -COOH + Cr (III) + H+ …….( Eq.2)

Cr2O72- + Fe2+(s) + H2O  =  Cr3+ + OH- + 6Fe3+(s) …….( Eq.3)

Surface C (+) + Cr2O72- = Surface C - Cr2O7 (Surface)…….( Eq.4)

And this was added in the SI

Scheme S4. Expected binding mechanism of chromate ions with MC-TDP sorbent at acidic pH medium

  1. Please indicate clearly why modification of chitosan devoted to sorption processes is needed. The introduction: the "second" part dedicated to chitosan needs explanation.
  2. The introduction should provide the knowledge/background for the aim of this work. In its current form, it is not enough. How and why can be the proposed materials better in Cr sorption than others? What problem will be solved/benefits? And finally – the novelty should be clearly seen. The main question is why such a modification has been proposed? Why were magnetic particles prepared?

Thanks for your comment, this part was modified and improved, we wish that the new version of manuscript meet your satisfaction

  1. In the whole manuscript, the results are given and described; however, there is a lack of any discussion – on why these changes occur, how these changes influence the material properties (are these changes positive or negative), and how they can affect the sorption properties of the material

Thanks for alerting, we try best to improve the discussion in the whole manuscript.

  1. Conclusions should also be improved. Why does the modification improve the sorption properties?

 Thanks, we try to improve the conclusion part and we wish it is fine in the recent state.

 Line 25: explain all abbreviations when they are used for the first time (e.g., FTIR, BET, etc.)

Thanks, it was revised and mentioned in the abstract

Line 25-29: sentence need revision (style)

Thanks, it was corrected

Line 29: what kind of conditions?

Thanks for alerting, it was described    

Line 29: "high sorption uptake" – high Cr(VI) sorption?

Thanks, it was corrected

Line 41” "suchlike" – should be "such like"

Thanks, it was corrected

Line 54: what is the limit?

Thanks, it was added

Line 62: not "absorbtion" but "adsorption – also check the other parts of the manuscript

Thanks for hard revision, it was corrected

Line 70 – to general sentence. What does "stability" means?

Thanks, it was described as stability (chemical and thermal stability)

Line 95: as the Authors stated that DA of chitosan is an essential parameter for its properties, the exact value of DA should be determined

Thanks for your comment but this description was performed by the manufactured company  

Line 130: 'Mg-chitosan" suggest not magnetic but chitosan with Mg element – consider other abbreviation

Thanks, it was corrected

FTIR analysis; please shortly describe, based on the observed changes, what are the main results regarding the sorption and desorption process,

Thanks, it was added

Line 227: not possible that the evaporation of water occurs at 270.8 C.deg (see DTG), this is the end of evaporation, however, max should be given (or better, onset temperature)

Thanks, the reviewer is correct, but in some modified crosslinking materials, some peaks were shifted toward higher temperature values. According to the literature survey, the crosslinked or modified materials are realizing water at high temperature than the low thermally stable or volatile one.  Reddy et al [1], assigned the water release of alginate based on silver nanocomposite hydrogels at 230°C, while Lee et al [2] reports the water release from neat cellulose acetate (CA) and CA/ Ni(NO3)2·6H2O matrix polymers ranged from 200°C to 300°C depending on the process used. Others assigned the absorbed water release from volatile compounds below 200°C [3] 

[1] Reddy, P., K. Madhusudana Rao, K. S. V. Rao, Yury Shchipunov, and Chang-Sik Ha. "Synthesis of alginate based silver nanocomposite hydrogels for biomedical applications." Macromolecular Research 22, no. 8 (2014): 832-842.

[2] Lee, Woong Gi, Do Hyeong Kim, Woo Cheol Jeon, Sang Kyu Kwak, Seok Ju Kang, and Sang Wook Kang. "Facile control of nanoporosity in Cellulose Acetate using Nickel (II) nitrate additive and water pressure treatment for highly efficient battery gel separators." Scientific reports 7, no. 1 (2017): 1-9.

[3] Ismail, N. H., W. N. W. Salleh, N. Sazali, and A. F. Ismail. "Effect of intermediate layer on gas separation performance of disk supported carbon membrane." Separation Science and Technology 52, no. 13 (2017): 2137-2149.

Line 264: Check the chemical formulas

Thanks, it was revised

Line 348: are the numbers in % or in the difference in sorption (indicate the unit)

Thanks, it is %

Section 2.3 – explain SD

Thanks, it was described

Figure 8: explain axis notation (SC, R%, SD)

 Thanks, it was mentioned

Check all parts f the manuscript: be sure to write "Cr(VI)" not "Cr (VI)".

Thanks, it was corrected in the whole MNs

English: "reaction maintained stirring" (line 121), "separated" (instead of "separation", line 125) etc

Mirror mistakes: line 90 (N2), line 184 (cm-1), line 190 (pikes), line 237 (DrTG), line 232 (is closed to) etc. – please carefully read the whole manuscript.

Thanks, we revised the whole manuscript and try our best to correct the mistakes

There are two schemes S1 in supplementary materials; see TDPD in Table S2 (lack of TDPD explanation, what is the difference between TDPD and TDP).

Sorry for this error, we correct it

We hope the revised version is now more understandable and meeting your satisfactions

We hope you will agree this revised version; however, we are ready for making any changes that  the Reviewer would consider useful and necessary.

Round 2

Reviewer 2 Report

The authors did not reply properly to reviewer comments:

  1. The authors discuss the use of modified chitosan for the removal of metal ions from aqueous effluents. For a practical application, it is not a cost-effective approach (the synthesis scheme of this work is expensive). There are other polymers or adsorbents of lower cost and high efficiencies for metal ion removal (a) Sun, Gang, and Weixing Shi. "Sunflower stalks as adsorbents for the removal of metal ions from wastewater." Industrial & engineering chemistry research 37, no. 4 (1998): 1324-1328.; b) Qasem, Naef AA, Ramy H. Mohammed, and Dahiru U. Lawal. "Removal of heavy metal ions from wastewater: A comprehensive and critical review." NJ Clean Water 4, no. 1 (2021): 1-15.)

Thanks for alerting; Actually, the work is based on the biopolymer (safe to the environment and safe at the end-of-life cycles) the main target of this work is to synthesis a highly removal Cr and efficient apply to the highly contaminated solution with improving the kinetic efficiency. The Sun, Gang, and Weixing Shi work have capacity toward Cr of around 25.07 mgg-1 comparing to 301 mgg-1 of our work and this is the most advantage as well as the reuses and stability.

The proposed method is a not cost-effective approach.

2. Magnetite DISSOLVES IN ACIDIC MEDIUM!!!!

The corresponding author refers the reviewer to previous publications of his research group to justify synthesis conditions of polymer magnetite composites. The author lists at least 8 articles regarding the use of polymer magnetite composites for the removal of metal ions in contaminated effluents. The publications are similar decreasing the novelty of the present study.

Thus, if the study is not suitable for practical application (COST) and does not add new knowledge, why should be considered for publication?

Author Response

Response to Reviewers

Red: Response to reviewer’s comment

Blue: New information added to the revised version of the manuscript.

Open Review

English language and style

(x) Extensive editing of English language and style required
( ) Moderate English changes required
( ) English language and style are fine/minor spell check required
( ) I don't feel qualified to judge about the English language and style

Yes

Can be improved

Must be improved

Not applicable

Does the introduction provide sufficient background and include all relevant references?

( )

( )

(x)

( )

Are all the cited references relevant to the research?

( )

( )

(x)

( )

Is the research design appropriate?

( )

(x)

( )

( )

Are the methods adequately described?

( )

( )

(x)

( )

Are the results clearly presented?

( )

(x)

( )

( )

Are the conclusions supported by the results?

( )

(x)

( )

( )

Comments and Suggestions for Authors

The authors did not reply properly to reviewer comments:

  1. The authors discuss the use of modified chitosan for the removal of metal ions from aqueous effluents. For a practical application, it is not a cost-effective approach (the synthesis scheme of this work is expensive). There are other polymers or adsorbents of lower cost and high efficiencies for metal ion removal (a) Sun, Gang, and Weixing Shi. "Sunflower stalks as adsorbents for the removal of metal ions from wastewater." Industrial & engineering chemistry research 37, no. 4 (1998): 1324-1328.; b) Qasem, Naef AA, Ramy H. Mohammed, and Dahiru U. Lawal. "Removal of heavy metal ions from wastewater: A comprehensive and critical review." NJ Clean Water 4, no. 1 (2021): 1-15.)

 Thanks for alerting; Actually, the work is based on the biopolymer (safe to the environment and safe at the end-of-life cycles) the main target of this work is to synthesis a highly removal Cr and efficient apply to the highly contaminated solution with improving the kinetic efficiency. The Sun, Gang, and Weixing Shi work have capacity toward Cr of around 25.07 mgg-1 comparing to 301 mgg-1 of our work and this is the most advantage as well as the reuses and stability.

The proposed method is a not cost-effective approach.

We fully understand the meaningful request of the Reviewer. To be honest the evaluation of cost statements requires competences not available in our group. Cost analysis (including reagent costs, operating/personal costs, etc. in addition to the accessible quantification of reagent costs, etc.) needs specific skills. Another critical criterion concerns the true cost for large-scale application: the industrial costs of reagents, resins is different to the costs available at lab-scale. For these reasons, we will consider this in the future work and support our group by these competences.

For these reasons, despite trying to give the best, we cannot provide a true comparison of the processes. Hence, we added to the Supplementary Information a new section including: the evaluation of the production costs for the sorbent, only based on reagent costs as shown below.

Table S2. Cost evaluations of 10 g MC-TDP sorbent for treatment of Cr(VI) ions

Chemicals used

Available units and prices

Units/prices for 10 g sorbent

Unit

Price (Euro)

Unit

Price (Euro)

Synthesis of TDP

Diethyl malonate

500 g

49

3.75 g

0.367

Thiourea

1000 g

121

1.5 g

0.1815

Sodium ethoxide

500 g

74

50 g

7.4

Synthesis of magnetite

Ferrous sulfate

250 g

30

4.166 g

0.4999

ammonium ferric (III) sulfate dodecahydrate

500 g

61

6.125 g

0.747

acetic acid

1000 mL

27

9 mL

0.243

Synthesis of modified sorbent

Chitosan

50 g

72

3 g

4.32

EPI

1000 mL

60

17 mL

1.02

Ethanol

2500 mL

61

50 mL

1.22

DMF

2500 mL

51

140 mL

2.856

Acetone

1000 mL

21

30 mL

0.63

Overall prices for 10 g sorbent

19.4844 euro

Notes: chemical reagents are analytical grade; the sorbent can be readily reused (see manuscript), and the experimental synthesis conditions are not optimized (meaning excess of reagents, which could be partly recycled). The operating costs for synthesis are not taken into account.

  1. Magnetite DISSOLVES IN ACIDIC MEDIUM!!!!

The corresponding author refers the reviewer to previous publications of his research group to justify synthesis conditions of polymer magnetite composites. The author lists at least 8 articles regarding the use of polymer magnetite composites for the removal of metal ions in contaminated effluents. The publications are similar decreasing the novelty of the present study.

As you see most of our publication was performed in a high-level journals that respect the novelty of the work as well as all of these articles differ in the final product that be used and the application. We add these references to appeared that we justified the base items of the synthesis. Also from the synthesized procedure the complete precipitation of Fe3O4 was performed at pH around 10, so if any percent that disoolved in the acidic medium during dissolving the chitosan it will be precipitated in the next step of producing the nano magnetite particles as reported by Massart method [1]. To verify that the magnetite was grafted by chitosan particles,  we make the EDX analysis for both sorbents that appeared the high percent of Fe that indicate that not degradation during synthesis.   

[1] Massart, R. Preparation of aqueous magnetic liquids in alkaline and acidic media. IEEE Trans. Magn. 1981, 17, 1247-1249.

This section was added

Figure 4 represents the semiquantitative EDX analysis for both sorbents (MCH and MC-TDP). The nonmodified sorbent shows low percent of O and N than that of grafted heterocyclic sorbent, on the other hand the sulfur was appeared in the grafted sorbent confirms the successive grafting of the TDP moiety. It was noteworthy that the presence of Fe ion verifies the stability of magnetite and nondegradable during synthesis, also decreasing Fe % in the MC-TDP than that measured in MCH is due to the relative in-creasing of the hydrocarbon constituents.

Figure 4. EDX analysis of MCH and MC-TDP

Thus, if the study is not suitable for practical application (COST) and does not add new knowledge, why should be considered for publication?

We hope you understand our consideration for the calculation of the cost, and we will consider this mission by supporting this calculation in the future work.

Reviewer 3 Report

The discussion of results (as I asked in my first review) was not improved.

Line 376: correct the sentence – it seems like being cut

Line 379: "the interaction was performed" – correct this statement

Line 380: protonated NH or NH2?

Line 382: the statement "N+" is too general. Something like this doesn't exist

Equations 1-4 must be corrected – the number of hydrogen/Fe/Cr atoms should be corrected – in general, both sides of the equation should have a uniform number of particular atoms, and this same sum of charge

What means 's' in equation 3?

Line 251-252: I agree with the explanation of higher temperatures (than 100 C.deg) of water evaporation however, I still do not agree with the value given. Water evaporation temperature is assigned to the "end" temperature (270.8). It is generally assumed that the onset-start or maximum DTG temperature should be given (or the whole transition range instead). Evaporation of water was not noticed at 270.8 but ended at this temperature.

Line 29: remove "wich"

Line 32: "fast kinetics of the sorbent" remove "of the sorbent" as fast  kinetics is a feature of sorption process, not a sorbent itself

What does the phrase mean: "which related to the dense of amine groups"? (incorrect word)

Still, several sentences need revision, like "Dissolving of ammonium ferric sulfate ( 7.35 g) is undergoes mixed with ferrous sulfate (5.0 g) in distilled water at 50 °C"

Author Response

Response to Reviewers

Red: Response to reviewer’s comment

Blue: New information added to the revised version of the manuscript.

Open Review

English language and style

( ) Extensive editing of English language and style required
(x) Moderate English changes required
( ) English language and style are fine/minor spell check required
( ) I don't feel qualified to judge about the English language and style

Yes

Can be improved

Must be improved

Not applicable

Does the introduction provide sufficient background and include all relevant references?

(x)

( )

( )

( )

Are all the cited references relevant to the research?

(x)

( )

( )

( )

Is the research design appropriate?

(x)

( )

( )

( )

Are the methods adequately described?

( )

(x)

( )

( )

Are the results clearly presented?

( )

( )

(x)

( )

Are the conclusions supported by the results?

( )

( )

(x)

( )

Comments and Suggestions for Authors

The discussion of results (as I asked in my first review) was not improved.

Thanks, we try best to improve the discussion section

Line 376: correct the sentence – it seems like being cut

Thanks, it was corrected

Line 379: "the interaction was performed" – correct this statement

Thanks, it was corrected

Line 380: protonated NH or NH2?

Actually the ratio of NH is higher than that of NH2, this is due to substitution reaction that performed during grafting of the heterocyclic moieties on the free NH2 of the chitosan moieties.

Line 382: the statement "N+" is too general. Something like this doesn't exist

Thanks, It was clarified

Equations 1-4 must be corrected – the number of hydrogen/Fe/Cr atoms should be corrected – in general, both sides of the equation should have a uniform number of particular atoms, and this same sum of charge

Thanks, it was corrected

What means 's' in equation 3?

S means “solid” and was mentioned

Line 251-252: I agree with the explanation of higher temperatures (than 100 C.deg) of water evaporation however, I still do not agree with the value given. Water evaporation temperature is assigned to the "end" temperature (270.8). It is generally assumed that the onset-start or maximum DTG temperature should be given (or the whole transition range instead). Evaporation of water was not noticed at 270.8 but ended at this temperature.

You is right, but still there is no typical identical on both sides of the TGA and DTG, this deviation is due to the model of the equipment and condition to the prepared sample

Line 29: remove "wich"

Thanks it was removed

Line 32: "fast kinetics of the sorbent" remove "of the sorbent" as fast  kinetics is a feature of sorption process, not a sorbent itself

Thanks, it was corrected

What does the phrase mean: "which related to the dense of amine groups"? (incorrect word)

It was corrected, the word dense used for describing the huge number of functional groups.

The phrase corrected to

which related to the high percent amine groups

Still, several sentences need revision, like "Dissolving of ammonium ferric sulfate (7.35 g) is undergoes mixed with ferrous sulfate (5.0 g) in distilled water at 50 °C"

Thanks, the whole manuscript was revised and hope meeting of your satisfaction 
